# Neonatal Screening in Europe Revisited: An ISNS Perspective on the Current State and Developments Since 2010

**DOI:** 10.3390/ijns7010015

**Published:** 2021-03-05

**Authors:** J. Gerard Loeber, Dimitris Platis, Rolf H. Zetterström, Shlomo Almashanu, François Boemer, James R. Bonham, Patricia Borde, Ian Brincat, David Cheillan, Eugenie Dekkers, Dobry Dimitrov, Ralph Fingerhut, Leifur Franzson, Urh Groselj, David Hougaard, Maria Knapkova, Mirjana Kocova, Vjosa Kotori, Viktor Kozich, Anastasiia Kremezna, Riikka Kurkijärvi, Giancarlo La Marca, Ruth Mikelsaar, Tatjana Milenkovic, Vyacheslav Mitkin, Florentina Moldovanu, Uta Ceglarek, Loretta O’Grady, Mariusz Oltarzewski, Rolf D. Pettersen, Danijela Ramadza, Damilya Salimbayeva, Mira Samardzic, Markhabo Shamsiddinova, Jurgita Songailiené, Ildiko Szatmari, Nazi Tabatadze, Basak Tezel, Alma Toromanovic, Irina Tovmasyan, Natalia Usurelu, Parsla Vevere, Laura Vilarinho, Marios Vogazianos, Raquel Yahyaoui, Maximilian Zeyda, Peter C.J.I. Schielen

**Affiliations:** 1International Society for Neonatal Screening (ISNS) Office, 3721CK Bilthoven, The Netherlands; peter.schielen@gmail.com; 2Department of Newborn Screening, Institute of Child Health, 11527 Athens, Greece; dplatis@ich.gr; 3Centre for Inherited Metabolic Diseases, Karolinska University Hospital and Department of Molecular Medicine and Surgery, Karolinska Institute, SE-17 76 Stockholm, Sweden; rolf.zetterstrom@sll.se; 4Newborn Screening Laboratories, Tel-HaShomer, 52621 Ramat Gan, Israel; shlomo.almashanu@sheba.health.gov.il; 5CHU-Domaine du Sart Tilman, 4000 Liège, Belgium; f.boemer@chuliege.be; 6Sheffield Children’s NHS Foundation Trust, Sheffield S10 2TH, UK; j.bonham@nhs.net; 7Laboratoire National de Santé, 3555 Dudelange, Luxembourg; patricia.borde@lns.etat.lu; 8Mater Dei Hospital, Tal-Qroqq Msida, MSD2090 Msida, Malta; ian.brincat@gov.mt; 9Groupement Hospitalier Est, 69500 Bron, France; david.cheillan@chu-lyon.fr; 10Centre for Population Research, National Institue for Public Health and the Environment (RIVM), 3720BA Bilthoven, The Netherlands; eugenie.dekkers@rivm.nl; 11National Genetic Laboratory, Hospital Maichin Dom, 1431 Sofia, Bulgaria; s3d@abv.bg; 12Neonatal Screening Laboratory, Children’s Hospital, CH-8032 Zürich, Switzerland; ralph.fingerhut@gmail.com; 13Department Genetics & Molecular Medicine, Landspitali, Reykjavik 108, Iceland; leifurfr@landspitali.is; 14University Children’s Hospital, 1000 Ljubljana, Slovenia; urh.groselj@gmail.com; 15Staten Serum Institute, 2300 Copenhagen, Denmark; dh@ssi.dk; 16Newborn Screening Centre, Banska Bystrica 97401, Slovakia; maria.knapkova@dfnbb.sk; 17Medical Faculty, 1000 Skopje, North Macedonia; mirjanakocova@yahoo.com; 18University Clinical Centre, Pristina 10000, Kosovo; vjosakotori@hotmail.com; 19Department of Pediatrics and Inherited Metabolic Disorders, Charles University-First Faculty of Medicine and General University Hospital, Prague 12808, Czech Republic; viktor.kozich@lf1.cuni.cz; 20Clinical and Diagnostic Center “Pharmbiotest”, LLC 93000 Rubizhne, Ukraine; kremezna@pharmbiotest.com.ua; 21Newborn Screening Centre, Turku University Hospital, 20521 Turku, Finland; Riikka.Kurkijarvi@tyks.fi; 22Meyer Hospital, 50139 Florence, Italy; g.lamarca@meyer.it; 23Medical Faculty, University of Tartu, 50411 Tart, Estonia; ruth.mikelsaar@ut.ee; 24Mother and Child Health Care Institute of Serbia, Belgrade 11070, Serbia; tanjamil5e@gmail.com; 25Neonatal Screening Center, 115580 Moscow, Russia; mitkinsl@rambler.ru; 26National Institute for Mother and Child Health, 050474 Bucharest, Romania; florentina.moldovanu@gmail.com; 27University Clinic, 04103 Leipzig, Germany; Uta.Ceglarek@medizin.uni-leipzig.de; 28Newborn Blood Spot Screening Laboratory, Dublin 1, Ireland; loretta.ogrady@cuh.ie; 29Institute of Mother and Child, 01-211 Warsaw, Poland; oltarzewski@op.pl; 30Norwegian National Unit for Newborn Screening, 0424 Oslo, Norway; rdpetter@ous-hf.no; 31University Hospital Medical Centre Zagreb, 10000 Zagreb, Croatia; dramadza@gmail.com; 32Republican Scientific Centre for Gynaecology and Perinatology, Almaty 050020, Kazakhstan; Sdamilya@mail.ru; 33Institute for Sick Children, 81000 Podgorica, Montenegro; samardzic@t-com.me; 34Republican Center Mother and Child Screening, Tashkent 100164, Uzbekistan; markhabo2@mail.ru; 35Centre for Medical Genetics, 08661 Vilnius, Lithuania; jurgita.songailiene@santa.lt; 36Children’s Clinic, 1083 Budapest, Hungary; szatmari.ildiko@med.semmelweis-univ.hu; 37NeugoGenetic and Metabolic Center, Tbilisi 0194, Georgia; n_tabatadze@hotmail.com; 38Child and Adolescent Health Department, 06430 Ankara, Turkey; Basak.Tezel@saglik.gov.tr; 39Department of Pediatrics, University Clinical Centre, Tuzla 75000, Bosnia and Herzegovina; almatoromanovic88@gmail.com; 40Arbes Health Care Centre, Yerevan 0014, Armenia; irinatovmasyan@yahoo.com; 41National Centre Health and Reproductive & Medical Genetics, 2062 Chisinau, Moldova; natalia.usurelu@yahoo.com; 42Children’s University Hospital, 1004 Riga, Latvia; parslavevere@inbox.lv; 43National Institute of Health, 4000-055 Porto, Portugal; laura.vilarinho@insa.min-saude.pt; 44Center for Preventive Paediatrics, 3022 Limassol, Cyprus; vogazianos@cpp.org.cy; 45Málaga Regional University Hospital. Institute of Biomedical Research IBIMA, 29011 Málaga, Spain; raquelyahyaoui@gmail.com; 46Department of Pediatrics and Adolescent Medicine, 1090 Vienna, Austria; maximilian.zeyda@meduniwien.ac.at

**Keywords:** neonatal screening, newborn screening, congenital metabolic disorders, rare diseases, dried blood spot screening, congenital endocrine disorders, International Society for Neonatal Screening, ISNS, public health

## Abstract

Neonatal screening (NBS) was initiated in Europe during the 1960s with the screening for phenylketonuria. The panel of screened disorders (“conditions”) then gradually expanded, with a boost in the late 1990s with the introduction of tandem mass spectrometry (MS/MS), making it possible to screen for 40–50 conditions using a single blood spot. The most recent additions to screening programmes (screening for cystic fibrosis, severe combined immunodeficiency and spinal muscular atrophy) were assisted by or realised through the introduction of molecular technologies. For this survey, we collected data from 51 European countries. We report the developments between 2010 and 2020 and highlight the achievements reached with the progress made in this period. We also identify areas where further progress can be made, mainly by exchanging knowledge and learning from experiences in neighbouring countries. Between 2010 and 2020, most NBS programmes in geographical Europe matured considerably, both in terms of methodology (modernised) and with regard to the panel of conditions screened (expanded). These developments indicate that more collaboration in Europe through European organisations is gaining momentum. We can only accomplish the timely detection of newborn infants potentially suffering from one of the many rare diseases and take appropriate action by working together.

## 1. Introduction

Neonatal screening, called newborn screening (NBS) in some countries, was initiated in Europe during the 1960s, when small programmes to screen for phenylketonuria (PKU) started in, e.g., the UK [1]. The detection of the biomarker for the disease, phenylpyruvic acid in urine, was a first and necessary step for screening as early as the 1930s [2,3,4]. However, the major breakthrough was the development of a simple and cheap screening method for PKU using dried blood spots (DBS) on a special collection device of a blood sample taken from the heel of newborn infants by Guthrie [5]. A few years later, Dussault introduced the radio-immunochemical methods for measuring the concentrations of thyroid hormones in DBS to detect congenital hypothyroidism (CH), enabling a second disease to be detected in NBS [6]. Gradually, the panel of screened disorders (“conditions”) expanded, with a boost in the late 1990s and the first decade of the 21st century with the introduction of tandem mass spectrometry (MS/MS) in NBS, making it possible to screen for 40–50 conditions using a single blood spot [7,8,9]. The most recent additions to screening programmes (screening for cystic fibrosis (CF), severe combined immunodeficiency (SCID) and spinal muscular atrophy (SMA)) were assisted by or realised through the introduction of molecular technologies. The Wilson and Jungner criteria [10] were the primary road map toward designing a screening policy taking into consideration prevalence, diagnostics, treatability, etc. However, huge advances in molecular technologies and other analytical innovations have allowed policymakers to include an ever-expanding list of disorders in their national programmes, sometimes stretching the intent of the criteria.

In the context of this paper, Europe is a geographical area consisting of around 50 countries situated east of the Atlantic Ocean, north of or in the Mediterranean Sea and west of the Ural Mountains, including all of Russia. In addition, the International Society for Neonatal Screening (ISNS) has decided to regard five former USSR republics, positioned east of the Ural (Kazakhstan, Kyrgyzstan, Tajikistan, Turkmenistan and Uzbekistan), to be part of Europe in view of their historically close ties to Russia [11]. Similarly, the ISNS has recognised Israel to be part of Europe following their request. In 2019, the total population of these countries was around 915 million, with an average annual birth rate of around 11.9 per 1000 people, leading to some 10.9 million newborn infants [12,13].

In Europe, NBS was introduced in the western part of the continent during the 1960s, spreading to the eastern part in the course of the next four decades (Figure 1). In virtually all countries, there is now some kind of institutionalised neonatal screening. Some countries are so small that screening is performed in a larger neighbouring country (Liechtenstein covered by Switzerland, Andorra and Monaco by France, San Marino by Italy, Kosovo partly by Serbia). In Albania, most of Kosovo and Tajikistan, there is no official NBS programme yet, but in Albania, there are local initiatives in some hospitals.

Although the value of NBS has been widely recognised, its introduction, depending on health care structure, available funds, local politics, input from professional groups and the general public, has led to quite varying approaches in the way these programmes have been set up, financed and governed. To obtain more information on these differences, an online survey, commissioned by the European Union to a European project team, was compiled in 2010 in which all aspects of screening programmes were addressed [14]. The survey covered the EU member states, (potential) candidate member states and European Free Trade Association (EFTA countries), totalling 38 countries. Results showed large variations in the panel of screened conditions (ranging from none to over 30 conditions), in specimen collection time after birth, screening methodology and storage time of residual specimen material (varying from a few months to ‘indefinite’). In addition, there were considerable differences in the process of confirmatory diagnostics, treatment and follow-up of screening results. In 2011, the project group, based on this inventory, provided a list of 60 recommendations to the EU Commission [15], but so far, none of them have been taken up by the EU. On the other hand, the EU has stimulated the formation of European Reference Networks (ERNs), i.e., collaborations of institutes in member states with aligned research and diagnostic aims, particularly in the area of Rare Diseases, with the aim of providing greater equity of access to centres of excellence for patients in Europe. The ERNs are focused on both diagnosis and treatment. The inclusion of neonatal screening seems evident, but the attention of the ERNs for neonatal screening could still be improved [16].

The initial initiative in 2010 to make an inventory of the state of neonatal screening in Europe has led to an informal network of colleagues, comprising nearly all members of the ISNS, who have been asked to update their data biennially, most recently in 2020.

In this paper, we report the developments between 2010 and 2020 and highlight the progress made in this period. We also identify areas where considerable progress can still be made, mainly by exchanging knowledge and learning from experiences in neighbouring countries. Smaller parts of the results have been published elsewhere [17,18,19,20,21].

## 2. Materials and Methods

### 2.1. Scope of the Survey

For most of the countries, data were collected from local members of the ISNS who are directly involved in their national NBS programme and regarded as being knowledgeable on all major aspects. When a country lacked the presence of an ISNS member, we tried to establish contacts through personal relations between ISNS members and local screening laboratories, e.g., in the former USSR republics.

### 2.2. Data Collection

Data were gathered using a standard form during the period 2012–2018. A slightly different questionnaire was used to obtain additional data in 2020, which was disseminated to the same countries in light of the preparation of yearly regional reports assembled for the ISNS. Questions pertained to all aspects of screening, with special reference to the analytical phase. The elements of the questionnaires are listed in Appendix A.

In several countries, neonatal screening programmes also include screening for congenital critical heart disease and for hearing disabilities. This paper is limited to conditions identified by screening using neonatal blood samples.

## 3. Results and Discussion

Data were received for 51 countries. Table 1 is an overview of some general characteristics concerning the population screened, as well as some logistical information concerning the NBS programmes. Table 2 and Table 3 highlight the conditions that are part of the screening panel.

### 3.1. NBS Infrastructure

To determine the state of the NBS system, basic information on the infrastructure is necessary. This information includes data on birth, preanalytic data (mode of sampling, sample transport, laboratory organisation, analytical procedures, reporting of results and clinical partners) and diagnostic and clinical follow-up of results. It is important to realise that countries have their own distinguished manner of organising their health care system and that this also applies to the neonatal screening system.

In Table 1, the results are presented per country. In some countries, however, the NBS programme is carried out under the responsibility of autonomous parts of the country, such as in Belgium, Bosnia-Herzegovina, Germany, Italy, Spain and the United Kingdom. For a few countries, no data or limited data were available.

#### 3.1.1. Number of Screening Laboratories and Average Annual Workload

Most countries have only one or two screening laboratories. Countries with a larger population generally have more laboratories, with Russia having 78 laboratories. There is no obvious relationship between the number of laboratories and the number of births, leading to varying annual workloads per laboratory. Due to changes in the structure of the neonatal screening programmes in recent years, the number of laboratories in France and Italy has decreased substantially. Finland introduced DBS screening in 2015 and has considered abandoning screening using cord blood. Also, Malta recently decided to switch from cord blood screening to DBS screening.

It is not easy to provide an optimal average number of samples per laboratory [14]. Of course, the input of labour and the use of the laboratory equipment should be as efficient as possible, which argues in favour of a high daily workload and, consequently, a low number of laboratories. A high daily workload would produce a more precise estimate of daily mean or median concentration for each analyte, and trends in measurements would present themselves timelier. This favours accurate daily quality assurance. It also makes it easier for the laboratory to obtain useful screening information on rare disorders. On the other hand, it can be argued that the screening process should be able to go on continuously with as few interruptions, e.g., by maintenance and technical failures, as possible. That would mean that all essential equipment would have to be available at least in duplicate. Therefore, it could be argued to include a higher number of closely collaborating laboratories, each with their own infrastructure, but with the possibility to provide backup for each other in case of emergencies or even calamities. The vast majority of countries are autonomous as far as newborn screening is concerned and do not want to rely on the resources of a neighbouring country. There is currently little cross-border collaboration for back-up laboratory facilities. However, Finland recently backed up Estonia when the latter had instrumentation problems.

In general, based on the data from Table 1, we can see that most countries with a total birth rate of around 100–20,000 neonates per year operate with a single screening laboratory, which is probably an efficient strategy. Under such conditions, it might be more economically sound, as well as scientifically beneficial, to use a single efficient operating unit to gather the entirety of data under a single managerial scheme. Above this birth rate, countries tend to have multiple screening laboratories. This may be related to the increased workload, but may also partly be influenced by various other factors, such as politico-geographical structure or other subdivisions of the country (e.g., Italy, Spain, France, Germany) and terrain and size (e.g., Russia). We also observed an opposite trend, where a higher-than-necessary number of labs are available (e.g., Serbia, Bosnia-Herzegovina), again possibly due to various socioeconomic or political factors. In general, an efficient design of the NBS network of a country will depend on a variety of factors that transcend scientific and efficiency considerations and consider the various other idiosyncrasies of each country.

#### 3.1.2. Coverage

In the majority of the countries, the coverage, defined as the percentage of newborns included in neonatal screening, is higher than 90%. In many countries, coverage is even higher than 99% despite the fact that NBS is not mandated in most countries, with Italy being an exception. Reported coverages of, e.g., 99.9% or 100% need to be regarded with caution. Even with the best birth registries and neonatal screening registries, a number of eligible neonates will be missed in screening, and it should also be remembered that the figures quoted for coverage will include families who decline the offer of screening in countries where this is optional. In Kyrgyzstan and Turkmenistan, the screening programmes started only recently. The initial coverage of 30% needs attention and may rise in the future. In general, coverage seems to be unrelated to the need of an informed consent for NBS or the programme being mandated.

#### 3.1.3. Screening Information, Parental Consent and Consent for Sample Storage

Most countries (94%) now have information materials for parents available either as written brochures or via websites. This is a significant improvement compared to 66% in 2010. Around 64% of the countries ask for consent, which is almost similar to the percentage in 2010. Consent for long-term storage of blood spot cards is requested in only 30% of countries (see Section 3.1.6.).

#### 3.1.4. Sample Logistics

Almost all countries screen using blood taken from the heel of neonates. The recommended time for sampling after birth varies per country, with 13% of countries recommending sampling after 24 h, 67% after 48 h, 18% after 72 h and 2% even later. These differences are mostly due to differences in the organisation of the screening programme and maternity care. Some countries sample in the hospital before mother and child are discharged. In other countries, e.g., in the UK, the sample is taken by the midwife or health visitor when the mother and child have returned home. At any rate, the median sampling window of 48–72 h after birth is (much) later than, e.g., in the USA.

Differences in sampling time may also reflect differences in the screening panel. For some conditions, the optimal window to measure the corresponding screening parameter is earlier than others. Screening for primary CH in early samples may be affected by elevated thyrotropin concentrations due to the physiological postpartum thyrotropin surge, leading to false-positive results. The opposite is true for the screening for MCAD, VLCAD and GA-I in samples taken later than 72 h after birth when concentrations of C8- and C14-acylcarnitine and glutarylcarnitine, respectively, have dropped following the stress of birth, which is associated with increased lipolysis [22,23,24,25]. Obviously, if the sampling time is very late (>120 h) for conditions such as CAH, GALT and many inborn errors of metabolism (IEMs), neonatal screening becomes increasingly inefficient because the affected neonate may present with clinical symptoms before the screening result is available.

The transition time (i.e., the time between sampling and analysis) pertains to the time needed to send the sample to the screening laboratory per regular mail or courier. In many countries, this takes 1–3 days, but in some cases, transition time can take up to 10 or more days. Unfortunately, for countries with longer transition times, there has been little improvement over the past decade.

Early sampling (<72 h, or preferably earlier) and a short transition time benefit timely confirmatory diagnostics and treatment.

#### 3.1.5. Analysis

Most laboratories have introduced manual or automated immunochemical assays. Tandem mass spectrometry (MS/MS) technology also has become a routine method in many countries. Croatia plans to introduce next-generation sequencing (NGS) technology as a second-tier for some conditions, which is already done in Norway to detect CF mutations and to detect mutations underlying many disorders in the programme [26].

#### 3.1.6. Storage of DBS Cards

After analysis, the blood spot cards are discarded or stored. There are large differences in policies concerning the length of storage, varying from a few months in Germany and Ukraine up to an indefinite number of years. Many countries (43%) store samples for at least 5 years, while a minority of countries store samples for 1–2 years or less.

There is an ongoing discussion regarding whether long-term storage of the dried blood cards is useful and, if so, for how long and under what storage conditions [27]. Arguments in favour include the possibility to retrospectively check the screening result when a child unexpectedly develops a condition and the possibility to determine the prevalence and feasibility of potential additions to the panel. Arguments against long-term storage concern the traceability to personal data and the difficulty in managing long-term storage in conditioned holdings. Therefore, not only safety aspects, but also pragmatic and privacy-related aspects, come into play. In 2018, the General Data Protection Regulation (GDPR) became effective within European Union member states, requiring programmes to obtain permission from parents to store the card of their child [28]. The data shown in Table 1 indicate that only 8 of 27 EU countries require such consent even though it is now mandatory.

#### 3.1.7. Reporting of Screening Results

Several countries make all screening results available to parents either online (e.g., through a digital patient/parent portal) or by mail. Other countries only report screening results to the health care providers and the parents only are informed if an action is required, such as a referral or a request for a second sample.

### 3.2. Panel of Screened Conditions

In Table 2 and Table 3 the panels of conditions for neonatal screening in the European countries are presented. Whereas “x” denotes that this condition is screened for in a considerable proportion of the population, ”p” denotes that the condition is screened for either in a pilot project or as part of a regional screening panel.

Historically, PKU screening was introduced first, followed by CH screening. Around 2005–2010, CAH and CF were introduced in several countries. Around the same time, MS/MS was introduced, enabling the screening of tens of additional conditions, mainly inborn errors of metabolism. We use roughly this order to discuss the current state of screening panels in the European countries.

-All countries except Montenegro screen for PKU. Finland and Malta introduced PKU screening after the gradual transition from cord blood to DBS screening. Cord blood is still used for CH screening.

-All countries except Moldova screen for primary CH. The Netherlands has also been screening for central hypothyroidism since the 1990s via the primary measurement of total thyroxine, supplemented by thyrotropin and thyroxine-binding globulin for samples, with 20% and 5% lowest total thyroxine levels, respectively [29,30]. Malta performs the simultaneous measurement of thyroxine and thyrotropin in all samples, also enabling screening for central hypothyroidism.

-Currently, screening for CF and CAH is performed in roughly 50% of the countries, almost twice as many as compared to 2010.

-The introduction of MS/MS technology in many laboratories across Europe led to a considerable increase of screening for amino acidemias (e.g., MSUD, Tyr type 1, CIT type 1), organic acidemias (e.g., GA type 1, IVA, PA, MMA) and fatty acid oxidation disorders (e.g., MCAD, LCHAD, VLCAD).

-For other conditions detectable by MS/MS, the current state is less clear. The choice to screen often depends on the medical and technical knowledge and/or the personal interest of scientists, clinicians and public health colleagues involved in the decision-making process, together with the availability of funding. This can also be influenced by patient groups who may voice support for screening. Conversely, the recognition that the identification of mild or ambiguous phenotypes and the relatively low positive predictive value achievable for some conditions can cause stress for families and may discourage inclusion by policymakers in some countries. A few examples of conditions quite rarely screened for are MATI/III in Russia and SCAD in Hungary, Iceland, Italy and North Macedonia.

-GALK screening recently started in The Netherlands in 2020. Biotinidase deficiency screening will start in Spain in 2021. GAL screening will start in Denmark in 2021.

-SCID and xALD screening have gained growing attention in the past decade. Seven countries now screen for SCID routinely, whereas six other countries are running pilot projects or regional programmes. The Netherlands started national SCID screening in January 2021 and Ireland (using ADA) will likely start at the end of 2021. As for xALD screening, Iceland, The Netherlands and Slovenia have started pilot projects.

-Another recently started discussion concerns the inclusion of SMA in the screening panel [31,32]. Belgium and Slovakia have embarked on pilot screening projects. Germany will start nationwide SMA screening in 2021, and Spain will start a pilot screening in two regions. The Netherlands plans to implement SMA screening within 2 years.

-In Europe, the evidence for the inclusion of lysosomal storage disorders in the screening panel is regarded quite widely as insufficient. Some regions in Italy have started pilot screening projects for a panel of four lysosomal storage disorders (LSDs) [33]. The Netherlands will implement MPS-1 screening in 2021. Notably, the situation in the US is markedly different, as screening for various LSD is routinely performed in many US states.

In general, as evident by the previous data, the panel of screened disorders varies between different countries. A brief look is sufficient to reveal that budget and financial factors are not the primary causes of this heterogeneity, although they do play a role. Countries with similar economic status may present different screening panels, and these differences can sometimes be so extensive that it cannot be justified by endemic condition frequency. This is clearly not a straightforward matter, and these differences may underline existing local approaches to NBS by the corresponding expert panel in each country that decides these issues. In many cases, these differences reflect the culture of certain countries regarding being more or less ‘willing’ to embrace technological advancements, or they may reflect the quality of communication between the NBS performing agency and the governing agent. So, such an issue is rather difficult to analyse per se, but pointing out certain key points might help in their resolution.

### 3.3. Short-Term Programme Developments

In the 2020 survey, country representatives were also requested to list expected developments in their NBS programme, varying from analytical improvements and additions to the screening panel to more fundamental changes.

Conditions that may be added, for example, include CAH in Hungary and Italy, CF in Sweden, GAL in Estonia, SCID in Italy and SCD in Portugal. Cyprus, Greece and Lithuania hope to introduce MS/MS screening and hence will expand their panels considerably.

Of note is that a regional Belgium laboratory screens for haemoglobinopathies using MS/MS technology and that the UK is considering introducing this methodology. We considered the use of next-generation sequencing technology, reduced costs and greater access to MS/MS and the use of big data technology (e.g., the application of R4S/CLIR to evaluate screening results) as important changes [34,35,36,37]. Finland, Iceland, Norway and Sweden use CLIR for evaluation of individual screening results [26,38]. The Czech Republic uses the CLIR website to evaluate cut-off levels. Estonia is considering automating the exchange of data with the CLIR database, and the Netherlands and Denmark are investigating how to implement the use of CLIR in order to improve the positive predictive value of the programme.

The introduction of assays that determine cystic fibrosis transmembrane conductance regulator (CFTR) mutations in blood spot samples, e.g., by line blot assays, may have been the earliest introduction of molecular technology in the screening laboratory. Assays for SMA and SCID in Europe are mostly assays specially adapted for screening laboratories that only need the most modest laboratory adaptations and educated personnel to perform these assays within the screening laboratory. Introducing next-generation sequencing (NGS) is yet another step forward. In the 2020 survey, only a few laboratories reported any activity in this field, with the exception of Norway and the UK. Norway is an early adaptor to this methodology and already published an analysis of NGS for CF screening in 2016 [39]. Norway has also started to use this technology as second-tier tests for IEMs and SCID [26]. The UK has conducted a limited trial of the use of NGS as part of the screening algorithm for CF and the results are being evaluated.

### 3.4. Future Perspectives

Looking toward the future in Europe, we identified four areas with expected significant future developments, i.e., methodological developments, developments in logistics, continuous process and effect evaluation and closer collaboration concerning the organisation of NBS in European countries.

#### 3.4.1. Methodological Developments

Except for an obvious development in many laboratories using more biochemical second-tier testing, we identified three areas of technological expansion. One is the use of high- resolution mass spectrometry to build profiles of metabolites or compounds instead of detecting single compounds. Whereas there is some published proof of principle for this technique, there are no reports of European countries currently exploring this technology.

The second methodology is using combined demographic and analytical data to build a risk estimation for a given condition instead of predicting the concentration of a single metabolite and/or metabolite ratios with corresponding cut-off values [37] (see also the special issue in this journal). In theory, this would yield far superior performance of neonatal screening, but initiatives to use CLIR need to still take off in Europe. Two reasons for the delay are the lack of covariate data such as birth weight, gestational age, sex and sampling time in hours on the screening cards, and the feasibility of laboratory information systems to exchange data with CLIR-like big data facilities preserving patient data safety.

Then, there is NGS technology, which is very versatile and could develop into a technology that would potentially be able to screen for, e.g., SMA, CF, SCID and other monogenic diseases in one sample [26]. The expansion of NGS technology in Europe needs to be awaited. Molecular technology will claim its place in the screening laboratory. Simplified molecular technology, adapted to work in the high-throughput environment of the screening laboratory, has already been implemented in the European laboratories screening for CF, SCID and SMA. A careful analysis of the way in which this is used will be important to avoid adding to uncertainty when classifying individual NGS screening results.

#### 3.4.2. Logistical Developments

For many decades, the sample card with demographic details, put in a paper envelope and sent to the laboratory by regular mail or a courier, has been the norm. However, some screening programmes are now modernising this process. Instead of writing the demographic data on the card, this data can also be added in a neonatal screening portal, where the screener connects the barcoded or QR-coded sample card with the demographics. Obvious advantages are that the size of the card is not limiting for the amount of information that the laboratory requests; that the sampling card will be sent to the laboratory without identifying personal data, avoiding the risk of incidental release of privacy sensitive details in case a sample is intercepted; and that screeners can avoid clerical errors when entering written data in the laboratory information management system. Such a system may also facilitate a parent portal to report the screening results to parents. This may be especially useful in those countries that report all results to parents, as well as when the screening result is negative.

#### 3.4.3. Process and Effect Evaluation

One item that has not yet been surveyed at length is how NBS programmes establish performance indicators such as sensitivity, specificity and predictive values. We should attempt to compare and contrast conditions with respect to the extent to which the screening programmes have been effective in reducing morbidity or mortality for each of the conditions, represent value for money and have benefits that outweigh their detriments. Countries may have an extensive panel of conditions and a relatively small annual screening volume. It would then take many years to obtain sufficient information about the screening performance for each individual condition. Therefore, use of the large CLIR database and infrastructure is recommended [34]. In addition, close collaboration between the screening laboratory professionals, the metabolic paediatricians, primary care providers and, ideally, clinical epidemiologists, is essential to be able to compare the screening results and the short-term and long-term clinical outcome, preferably using a common database in which all results per infant are registered and evaluated periodically. In this way, a good overview of false-positives and false-negatives can be obtained, which may lead to further methodological improvements.

Another item for thought is the question of whether NBS programmes have set performance goals, such as increasing coverage (if applicable), increasing positive predictive values, laboratory time to result, minimising clerical errors and the follow-up of patients picked up by the programmes. European collaboration regarding the use of international databases and tools could further support programme improvement within Europe.

We realise that there are a lot of data to be scrutinised. Many screening tests are performed quite differently, e.g., CF screening can be performed in a one-one to four-tier procedure depending on the country where the screening is performed. For CAH screening, various algorithms exist in case a result is intermediate. A second sampling is often performed, and some laboratories have second-tier tests in place. Whereas it is clear that this level of detail is not within the scope of this study, an even more detailed state of affairs may be explored in future surveys. In particular, the performance indicators of programmes should be evaluated, as this would identify best practices. For this, immaculately organized follow-up is needed, although at least robust reporting of false-positive results could be a very good quality indicator. Given that these are all rare diseases, and that most countries will find none to a few in a decade, it is clear that European collaboration concerning the collection of data is very important.

#### 3.4.4. Closer Collaboration Concerning the Organisation of NBS across Europe

In spite of the consensus concerning the value of NBS, there are large differences among countries on how NBS is embedded in the health care systems, how the intake of infants is done, at what time after birth and by whom, how the parents are informed and sometimes asked for written consent, how parents and caretakers are informed about the screening results and the subsequent action for referral and treatment.

Each country is governed independently and makes its own decisions concerning conditions that should be included in NBS. Unlike the USA, where public opinions can influence NBS policies to a great extent, there is little public knowledge concerning the health care organisation in neighbouring countries. As a consequence, advocacy efforts concerning health policies across borders are limited.

In Europe, health care is usually funded through a national health service or a statutory health insurance (social security). In almost every jurisdiction, public health care systems have defined mechanisms to assess and appraise payment for NBS, and most allow for diverse stakeholder participation when considering technology effectiveness, disease severity, the health economy and treatment availability. This often results in complex governmental (financial) decisions when expansion to include new conditions is considered.

Table 2 and Table 3 show the large variations among countries. Since the previous study concerning the situation in 2010 [18], there have been major changes in the screening panels, largely due to the increased availability of MS/MS technology making it possible to obtain the results for 10–15 conditions in using single DBS sample. Yet, the appreciation of the value of screening for certain conditions differs and seems to depend, to a certain extent, on the available knowledge and interest of leading (para)medical opinion leaders.

The assessment of long-term outcomes for patients identified by screening can best be achieved by international collaboration. The disorders are rare, and the use of international disease registries founded upon agreed case definitions continues to be a pressing area where progress is required. Notable advances such as the European Registry and Network for Intoxication type Metabolic Diseases (E-IMD) and the European Registry and Network for Homocystinurias and Methylation Defects (E-HOD) offer encouragement. In addition, prioritization criteria for genetic testing services regarding medical benefits, health needs and cost have been proposed [40]

There are currently no policy recommendations for NBS, nor is there direct oversight at the European level or within the EU [16]. Over time, the EU, consisting of 27 countries as of 2021, has established several treaties on topics to be governed or overseen by the European Commission (EU’s Executive Body). However, health care has not been included because the member states consider it to be their own responsibility (principle of subsidiarity). To add complexity, in some EU countries, e.g., Belgium, Germany, Italy and Spain, as well as in the UK, policymaking is decentralized to regions or provinces that function more or less autonomously.

There may be some hope from the creation by the EU of the European Reference Networks for the diagnosis and treatment of rare diseases, including inherited metabolic disorders, immune deficiencies and some rare endocrine disorders. The screening community must emphasise that the patients involved are, for a large part, identified by NBS, and that appropriate screening programmes should form an integral part of these important networks as they develop [16].

## 4. Conclusions

Between 2010 and 2020, most NBS programmes in geographical Europe matured considerably, both in terms of methodology (modernised) and with regard to the panel of conditions screened (expanded). It must be noted that the results presented here are subject to new changes within a short time window. The ISNS is in the process of establishing a database which is available for public consultation [41].

This paper reports a cross-sectional study of European screening programmes. It should inspire the programmes to produce per-country disorder-based overviews to compare screening performance, diagnostic confirmation and therapeutical follow-up.

Personal contacts between and among programme leaders have been stimulated by the ISNS during conferences and workshops. On a European level, the ISNS recently established close links with the International Patient Organization for Primary Immunodeficiencies (IPOPI), the European Reference Network for Hereditary Metabolic Diseases and Rare Endocrine disorders (metabERN/EndoERN) and the European Organization for Rare Disease (EURORDIS). In addition, the ISNS, IPOPI and the European Society for Immunodeficiencies (ESID) are in the process of establishing a stakeholder platform for neonatal screening as part of the EU Health Policy Platform, called Screen4Rare.

Together, these developments indicate that more collaboration in Europe through European organisations is gaining momentum. Whether this will ultimately lead to, e.g., a European draft priority list for neonatal DBS screening for various conditions [15] to assist policymakers faced by the wide variation in the current practice of neonatal DBS screening within Europe, taking into account published international guidelines such as those by the ISNS [42], or whether Europe will choose another path toward more collaboration, remains to be seen.

We can only accomplish the timely detection of newborns potentially suffering from one of the many rare diseases and take appropriate action, and prepare for the implementation of new developments for all European citizens, by working together.

## Figures and Tables

**Figure 1 IJNS-07-00015-f001:**
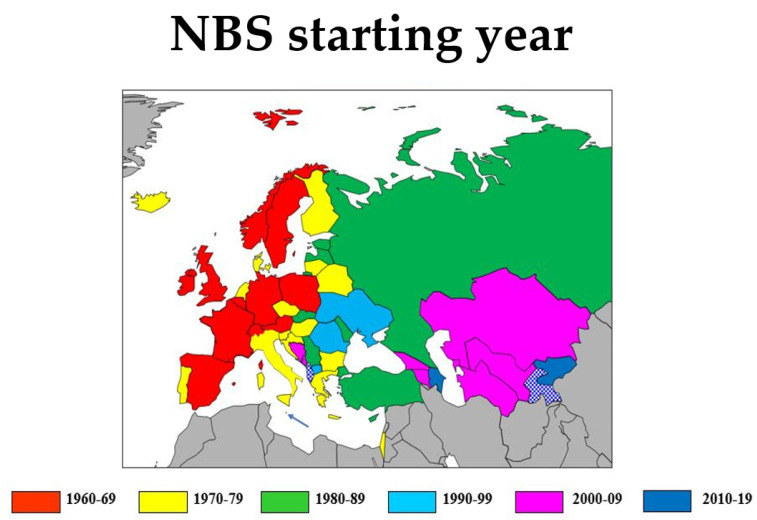
Map of the International Society for Neonatal (ISNS) European region, colour coded by the starting year of neonatal screening. Arrow indicates Malta (green). Albania and Tajikistan (in blue-hatched white) have no neonatal screening programme. Grey areas fall outside the geographical region under consideration. White areas are bodies of water.

**Table 1 IJNS-07-00015-t001:** Country data.

Country	Approx. Population 2020 (Million) ^1^	Approx. Number of Infants	Number Screening Laboratories	Average Number Samples Per Lab.^2^	Interval Birth-Sampling (hrs)	Interval Sampling-Analysis (Days)	% Infants Screened	Information to Parents Available?	Consent Participation?	Consent Storage?	Length Storage (yrs)	Normal Results Reported to Parents?
**Albania** (no screening)	3.0	36,000	n.a.^3^	n.a.	n.a.	n.a.	n.a.	n.a.	n.a	n.a	n.a	n.a.
**Austria**	8.8	87,000	1	87,000	36–72	1–3	>99.5	yes	no	no	10	no
**Armenia**	3.1	36,000	2 ^4^	36,000	48–96	1–5	99.8	yes	yes	no	I ndef.	no
**Azerbaijan**	9.7	170,000	1	35,000	48–72	3–5	30	yes	no	no	n.d.	n.d.
**Belarus**	9.8	108,000	1	108,000	72–120	1–5	n.d.	yes	no	no	5	n.d.
**Belgium**	10.5	117,000	4	30,000	48–120	2	99.8	yes	no	no	5	no
**Bosnia-Herzegovina**	3.3	28,000	3 ^4^	9000	48–96	1–7	96	no	no	no	10	no
**Bulgaria**	7.4	61,000	2 ^4^	61,000	72–120	5–10	n.d.	no	yes/no	no	20	n.d.
**Croatia**	4.2	36,200	1	36,200	48–72	3–5	100	yes	no	no	5	no
**Cyprus ^5^**	1.1	9500	1	9500	48–168	5–10	>99.9	yes	yes	no	2	no
**Czech Republic**	10.6	113,000	4 ^4^	56,000	48–72	2.5	100	yes	yes	no	5	no
**Denmark**	5.6	63,000	1	63,000	48–72	1–2	99.1	yes	yes	yes	indef.	online
**Estonia**	1.3	13,500	1	13,500	48–72	2–5	99.55	yes	yes	no	>25	no
**Finland**	5.5	45,000	1 ^4^	45,000	48–120	1–5	99	yes	yes	no	varies	online
**France**	67	760,000	16	47,000	48–72	2–3	99.96	yes	yes ^6^	no	1	no
**Georgia**	3.7	48,500	1	48,500	48–72	14–15	100	yes	no	no	15	no
**Germany**	80	787,000	11	71,000	36–72	2–3	100	yes	yes	yes	<1	no
**Greece**	10.5	80,000	1	89,000	48–72	6–8	100	yes	no	no	2	no
**Hungary**	10	90,000	2	50,000	48–72	3–4 *	99.99	yes	no	no	indef.	no
**Iceland**	0.35	4500	1	4500	48–72	3–5	100	yes	yes	yes	indef.	online
**Ireland**	4.9	59,700	1	59,700	72–120	1–2	>99.5	yes	yes	yes	10	no
**Israel**	9.2	194,000	1	194,000	36–72	1–3	99.8	yes	no	no	5	online
**Italy**	60.5	434,000	15	28,900	48–72	1–4	96.7	yes	no	no	2–10	no
**Kazakhstan**	18.7	402,000	21	20,000	24–72	1–2	96.5	yes	yes	no	3	no
**Kosovo**	1.8	25,000	n.a.	n.a.	n.a.	n.a.	n.a.	n.a.	n.a	n.a	n.a	n.a.
**Kyrgyzstan**	7.0	160,000	1	32,000	48–72	3–5	30	yes	no	no	n.d.	n.d.
**Latvia**	1.9	20,800	1	20,000	48–72	5–7	98.5	yes	yes	no	7	online
**Lithuania**	2.8	24,600	1	24,600	48–96	2–9	99.6	yes	yes	no	20	no
**Luxembourg**	0.6	7200	1	7200	48–72	4	>99	yes	yes	no	indef.	no
**Malta**	0.48	4400	2 ^4^	4400	72–120	5	99.7	yes	yes	no	indef.	no
**Moldova**	3.5	37,400	1	37,400	>48	30	92.3	yes	yes	no	0	n.d.
**Montenegro**	0.62	7200	1	7200	24–72	1–3	100	no	no	no	0.5	no
**Netherlands**	17.8	168,500	5	34,000	72–96	1–3	99.3	yes	yes	yes	5	mail
**North Macedonia**	2.1	20,000	1	20,000	32–72	3	>98	yes	no	yes	3	no
**Norway**	5.3	55,500	1	55,500	48–72	1–3	>99	yes	yes	yes	indef.	no
**Poland**	38.4	373,000	6	62,000	48–96	3	99.8	yes	yes	yes	5	online
**Portugal**	10.3	87,300	1	87,300	48–72	1–3	99.5	yes	yes	no	5	online
**Romania ^7^**	19.6	185,600	5	31,500	48–72	18	85	yes	yes	no	5	no
**Russia**	142	1,670,000	78 ^4^	20,000	48–72	3–5	90-92	yes	yes	no	3	n.d.
**Serbia**	7.0	65,000	2	32,500	48–72	3–5	99	yes	no	no	5	no
**Slovakia**	5.4	57,000	1	57,000	72–96	2–3	100	yes	yes	no	indef.	no
**Slovenia**	2.07	20,000	1	20,000	48–72	1–2	>99	yes	no	no	indef.	no
**Spain**	46.5	372,000	15	24,800	24–72	3–10	99.2	yes	yes	yes	5-indef	mail
**Sweden**	10	116,000	1	116,000	48–72	1–3	>99.5	yes	yes	yes	indef.	no
**Switzerland**	8.1	88,000	1	88,000	72–96	2	>99.9	yes	yes	yes	indef.	no
**Tajikistan** (no screening)	9.4	291,000	n.a.	n.a.	n.a.	n.a.	n.a.	n.a.	n.a	n.a	n.a	n.a.
**Turkey**	84.3	1,300,000	2	650,000	48–72*	1–2	97	yes	yes	yes	5	no
**Turkmenistan**	6.0	110,200	1	35,000	48–72	3–5	30	yes	no	no	n.d.	n.d.
**United Kingdom**	66.6	760,000	16	47,500	120	3–4	96.5	yes	yes	no	>5	mail
**Ukraine**	43.7	393,000	7	56,100	48–72	3	n.d.	yes	yes	n.a.	<0.5	mail
**Uzbekistan**	31.3	760,600	14	54,000	72–96	10.	95	yes	yes	n.d.	1	n.d.

Colour indications countries: EU member (brown), candidate EU member (fuchsia), potential candidate EU member (green), EFTA member (blue), Europe-other (purple).
^1^
https://worldpopulationreview.com/ (visited 14 November 2020); ^2^ in larger countries, numbers vary largely; ^3^ n.d. = no data; n.a. = not applicable; ^4^ not all laboratories carry out the whole screening panel; ^5^ data regarding northern part of the Cyprus island not included; ^6^ France only consented to cystic fibrosis transmembrane conductance regulator (CFTR) analysis; ^7^ Romania total newborns includes Romanian infants born elsewhere; % screened excludes private labs.

**Table 2 IJNS-07-00015-t002:** Screening panel, part 1.

	Endocrine Disorders			Amino Acidemias									Organic Acidemias								
	CH ^1^	CAH	CF	PKU	MSUD	HCY	Tyr-1	Tyr-2	ASA	Cit.1/2	ARG	MAT I/III	GA1	IVA	3MCC	PA	MMA ^3^	BKT	HCSD	3HMG	MCD
**Albania**																					
**Armenia**	x ^2^			x																	
**Austria**	x	x	x	x	x	x	x	x	x	x	x		x	x		x	x				
**Azerbaijan**	x	x		x																	
**Belarus**	x			x																	
**Belgium**	x	x	x	x	x	x	x	x					x	x		x	x				
**Bosnia-Her.**	x			x																	
**Bulgaria**	x	x		x																	
**Croatia**	x			x									x	x							
**Cyprus**	x			x																	
**Czech Rep.**	x	x	x	x	x	x				x	x		x	x							
**Denmark**	x	x	x	x	x		x		x				x	x		x	x				x
**Estonia**	x			x	x	x	x	x		x	x		x	x		x	x				
**Finland**	x	x		x	x	x	x		x	x	x		x	x		x	x				
**France**	x	x	x	x																	
**Georgia**	x		x	x																	
**Germany**	x	x	x	x	x		x						x	x							
**Greece**	x			x																	
**Hungary**	x		p ^1^	x	x	x	x	x	x	x			x	x	x	x	x	x		x	x
**Iceland**	x	x		x	x	x			x	x	x		x	x	x	x	x	x	x	x	x
**Ireland**	x		x	x	x	x							x								
**Israel**	x	x		x	x	x	x	x	x	x	x	x	x	x		x	x	x	x	x	x
**Italy**	x	p	x	x	x	x	x	x	x	x	x		x	x	x	x	x	x	x	x	x
**Kazakhstan**	x		p	x																	
**Kosovo**																					
**Kyrgyzstan**	x																				
**Latvia**	x	x	x	x																	
**Lithuania**	x	x		x																	
**Luxembourg**	x	x	x	x																	
**Malta**	x			x																	
**Moldova**				x																	
**Montenegro**	x		x																		
**Nethelands**	x	x	x	x	x		x						x	x	x	x	x			x	x
**N. Macedonia**	x		p	x	x	x	x		x	x	x		x	x	x	x	x	x	x		x
**Norway**	x	x	x	x	x	x	x						x	x		x	x	x	x	x	x
**Poland**	x	x	x	x	x	x	x	x	x	x	x		x	x	x	x	x	x		x	
**Portugal**	x		x	x	x	x	x	x	x	x	x		x	x	x	x	x			x	
**Romania**	x			x			x	x													
**Russia**	x	x	x	x								x									
**Serbia**	x		x	x												x					
**Slovakia**	x	x	x	x	x		x	x		x	x	x	x	x	x	x	x	x		x	
**Slovenia**	x			x	x		x						x	x	x	x	x	x	x	x	
**Spain**	x	p	x	x	p	p	p	p	p	p	p		x	p	p	p	p	p	p	p	p
**Sweden**	x	x		x	x	x	x		x	x	x		x	x		x	x	x			
**Switzerland**	x	x	x	x	x								x								
**Tajikistan**																					
**Turkey**	x	x	x	x																	
**Turkmenistan**	x	x	x	x																	
**United Kingdom**	x		x	x	x	x							x	x							
**Ukraine**	x	x	x	x	p	p	p	p	p	p	p		p	p	p		p	p	p		
**Uzbekistan**	x		p	x	p	p	p		p	p	p		p	p	p	p	p				
**Total ^4^**	**47**	**24 + 2**	**25 + 4**	**46**	**22 + 3**	**16 + 3**	**19 + 3**	**10 + 2**	**11 + 3**	**13 + 3**	**12 + 3**	**3**	**24 + 2**	**21 + 3**	**9 + 32**	**18 + 2**	**17 + 3**	**10 + 2**	**6 + 2**	**10 + 1**	**8 + 1**

Colour indications countries: EU member (brown), candidate EU member (fuchsia), potential candidate EU member (green), EFTA member (blue), Europe-other (purple).
^1^ for explanation of abbreviations, see the list in the main text; ^2^ x = in screening panel; p = pilot/regional screening; ^3^ total number of conditions in screening panel; total number of pilots; ^4^ MMA includes MMAmut, Cbl deficiencies and Vitamin B12-deficiency; Note North Macedonia 30% full tandem mass spectrometry (MS/MS) spectrum; Russia 20% full MS/MS spectrum.

**Table 3 IJNS-07-00015-t003:** Screening panels continued.

	Fatty acid Oxidation Disorders									Miscellaneous								Hemo	
	GA2 ^1^	MCAD	LCHAD/TFP	VLCAD	SCAD	CPT1	CPT2	CACT	CUD	RMD	GAL	BIOT	UDP	G6PD	xALD	SCID	SMA	SCD	Total ^3^
**Albania**																			**n.d. ^4^**
**Armenia**																			**2**
**Austria**	x ^2^	x	x	x		x	x	x	x	x	x	x							**26**
**Azerbaijan**											x			x					**5**
**Belarus**																			**2**
**Belgium**	x	x		x							x	x		p			p	p	**17 + 3**
**Bosnia-Herzegovina**																			**2**
**Bulgaria**																			**3**
**Croatia**		x	x	x					x										**8**
**Cyprus**																			**2**
**Czech Republic**		x	x	x		x	x	x		x		x							**18**
**Denmark**		x	x	x					x		x	x				x	p		**19 + 1**
**Estonia**		x	x	x		x	x	x	x										**19**
**Finland**	x	x	x	x		x	x	x	x							p			**21 + 1**
**France**		x																x	**6**
**Georgia**																			**3**
**Germany**		x	x	x		x	x	x			x	x				x	p	p	**17 + 2**
**Greece**											x			x					**4**
**Hungary**	x	x	x	x	x	x	x	x			x	x							**26 + 1**
**Iceland**	x	x	x	x	x	x	x	x	x						p	x			**27 + 1**
**Ireland**		x									x								**8**
**Israel**	x	x	x	x		x	x		x		x		x		p	x			**19 + 1**
**Italy**	x	x	x	x	x	x	x	x	x	x	x	x		p		p	p		**31 + 4**
**Kazakhstan**																			**2 + 1**
**Kosovo**																			**n.d.**
**Kyrgyzstan**																			**1**
**Latvia**											x	x							**6**
**Lithuania**											x								**4**
**Luxembourg**		x																	**5**
**Malta**																		x	**3**
**Moldova**																			**1**
**Montenegro**																			**2**
**Netherlands**		x	x	x		x					x	x			x	p		p	**20 + 2**
**North Macedonia**	x	x	x	x	x	x	x	x	x										**25 + 1**
**Norway**		x	x			x	x	x	x			x				x			**23**
**Poland**	x	x	x	x		x	x	x	x		p	x				p			**27 + 2**
**Portugal**	x	x	x	x		x	x	x	x										**24**
**Romania**																			**4**
**Russia**											x								**6**
**Serbia**																			**3**
**Slovakia**		x	x	x	x	x	x	x	x							p	p		**25 + 2**
**Slovenia**		x	x	x		x	x	x	x						p				**19 + 1**
**Spain**	p	x	x	p	p	p	p		p	p	p	p				p		x	**7 + 26**
**Sweden**	x	x	x	x		x	x	x	x		x	x				x			**25**
**Switzerland**		x									x	x				x			**10**
**Tajikistan**																			**n.d.**
**Turkey**												x							**5**
**Turkmenistan**											x			x					**6**
**United Kingdom**		x																x	**9**
**Ukraine**	p	p	p	p		p	p	p	p		p	p							**4 + 24**
**Uzbekistan**		p	p	p		p	p		p			p							**2 + 19**
**Total ^3^**	**11 + 2**	**26 + 2**	**20 + 2**	**19 + 3**	**+**	**17 + 3**	**16 + 3**	**15 + 1**	**15 + 3**	**+**	**17 + 3**	**13 + 3**	**1**	**3 + 2**	**1 + 3**	**7 + 6**	**0 + 5**	**4 + 3**	

Colour indications countries: EU member (brown), candidate EU member (fuchsia), potential candidate EU member (green), EFTA member (blue), Europe-other (purple).^1^ for explanation of abbreviations, see the list in the main text; ^2^ x = in screening panel; *p* = pilot/regional screening; ^3^ total number of conditions in screening panel; total number of pilots; ^4^ no data; Note North Macedonia 30% full MS/MS spectrum; Russia 20% full MS/MS spectrum.

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
