# Peer review of "Neonatal Screening in Europe Revisited: An ISNS Perspective on the Current State and Developments Since 2010"

_2409-515X, 2021, doi:10.3390/ijns7010015_

Round 1

Reviewer 1 Report

This is an excellent collection of data, working with a total of 41 authors to summarize the very important changes that have taken place in the newborn screening programs in Europe over the perios from 2010 to present.  The changes are impressive from many standpoints. Although there are many independent countries involved in these complex decisions about what to include in the newborn screening programs, it would be very helpful if there was an effort made to develop a plan by which decisions are made about how to decide what to include in newborn screening programs.  These plans will continue to differ by counries.

Author Response

Reviewer 1:

We thank reviewer 1 for the kind compliments. We hope that his wish that a plan be developed about how to decide what to include in newborn screening is reflected in the discussion of our manuscript, especially the final paragraph of section 5, and the additions made as per the comments of reviewer 3.

Reviewer 2 Report

Thank you very much for this outstanding manuscript.

This paper describes in detail NBS in Europe and its expansion. It is an essencial paper and hopefully after its publication other similar papers can be published about screening in other continents.

This paper is very well-written and it has a comprehensive amount of data in regards to NBS in Europe. It also mentions the Screen4Rare platform that will be very useful for NBS providers.

I suggest that the authors could improve the resolution of figure 1 (it is slightly pixelated).

Author Response

Reviewer 2:

We thank for the positive comments.

On this reviewers request, we provided a figure with higher resolution, also as per the comments of reviewer 3.

Reviewer 3 Report

This paper is a worthy effort to document which countries in WHO Europe are currently undertaking dried blood spot (DBS) neonatal screening and the medical disorders for which this screening is being undertaken. It is based on an informal  survey of International Neonatal Screening Society members between 2012 and 2018 including requests for updates sent every other year. The extent to which this survey is likely to provide accurate and complete information is not discussed (although I am not aware of any reason to doubt it).

There is wide variation between countries with respect to the number of conditions for which screening is undertaken or planned, as is apparent from the right hand column of Table 2b.  In Western Europe, for example, fewer than 10 conditions are screened for in Croatia, France, Greece and the UK while more than 30 conditions are screened for in Spain and Italy. This seems unlikely to represent a difference in budget that is potentially available for screening activity and some countries have national committees that has made  reasoned decisions to refrain from screening for some of the conditions listed.

While some of this variation may reflect differences in disease incidence, it seems likely that most of it is due to variation in the extent to which the Wilson and Jungner criteria have been ‘stretched’, as alluded to by the authors in the opening paragraph.

It would be very interesting to attempt to compare and contrast conditions with respect to the  extent to which the screening programmes have been effective in reducing morbidity or mortality for each of the conditions, represent value for money and have benefits that outweigh the harms. The paper would be improved by some discussion of these issues.

According to a morbidity and mortality report regarding screening in the USA published near the beginning of the 2010s, about 49 conditions were approved for neonatal screening (by DBS or by other means) in the USA at that time and 43% of all cases of all conditions were cases of permanent childhood hearing impairment. Screening for hip dysplasia and cataract are other examples of important neonatal screening not considered in this paper. I would therefore suggest inserting the phrase ‘dried blood spot’ within its title to indicate the limits to its scope.

Table 2 column headings are entirely comprised of abbreviations which should probably be explained in a footnote. Failing that, a Table footnote should refer the reader to the list of abbreviations provided at the end of the text.

Figure 1 is difficult to read and lat/long scales should be added to the axes. In the present plot, it looks as if the areas shown uncoloured on the left hand edge of the right hand box are shown coloured in the magnified left hand box which is confusing. The whole plot would be easier to interpret if a) the boundary of the area covered by the survey were explicitly marked and b) areas falling outside that boundary (e.g. Egypt) could be easily distinguishable (i)  from bodies of water (e.g. Caspian and Black Seas) and (ii) from  countries within the areas surveyed but lacking a neonatal DBS screening programme (e.g. Tajekistan).

Even better would be a single map plot of the whole of Europe at the lower magnification (at which the Eastern part is shown in the current right hand box) with a box around that part which is Western Europe linked by a line to a more highly magnified map of this area (as currently shown in the left hand box). This would avoid confusion about the areas shown on both of the current boxes.

Overall, the paper will provide useful information for those involved in documenting current DBS neonatal screening. It would be much more interesting if it could also venture into a discussion about the reasons behind the variations in screening programmes between countries and add discussion of the question of the relative merits and/or a draft priority list for neonatal DBS screening for various conditions to assist policy makers faced by this wide variation in the current practice of neonatal DBS screening within Europe.

Author Response

Reviewer 3.

We thank this reviewer for his valuable comments which we address below. We feel that reviewer 3 makes six valuable points.

1)Reviewer 3 observes that the extent to which this survey is likely to provide accurate and complete information is not discussed.

While we cannot guarantee that the information provided is accurate, the sources are all professionals in neonatal screening, co-authors of the manuscript and would indicate if their information was not valid. To stress this point, we amended the text of the M&M section.

Line 155-156: and regarded as being knowledgeable on all major aspects.

2)Reviewer 3 questions whether the differences in programmes represent a difference in budget that is potentially available for screening activity and some countries have national committees that has made  reasoned decisions to refrain from screening for some of the conditions listed. And that It would be very interesting to attempt to compare and contrast conditions with respect to the  extent to which the screening programmes have been effective in reducing morbidity or mortality for each of the conditions.

We agree with this reviewer that the extensive variety in programs, as described in this manuscript, immediately provokes thought on the causes of this variation. That is beyond the scope of this paper. To address the important point this reviewer makes, we added three paragraphs to the discussion.

Paragraph 3.2 last sentences. In general as evident …..towards their resolution.

Paragraph 3.4.3 second sentence. We should attempt to compare and contrast ….value for money and have benefits that outweigh the harms

5th sentence. In addition, close collaboration between ….. of false-positives and false-negatives can be obtained which may lead to further methodological improvements

3) Reviewer 3 points to the fact that neonatal screening may also comprise various forms of non blood spot related screening e.g. hip dysplasia etc. and that it should be indicated that these are not within the scope of this study.

We understand this suggestion. We feel that neonatal screening implicitly pertains to screening using neonatal blood and feel that to keep the title concise we should not change it at this time, as suggested by the reviewer. However, to clarify the scope of the paper and to address this reviewers comment, we stated the limitations of the scope of our paper more explicitly in the Materials and Methods section under b Data collection, last sentence.

In several countries neonatal screening programmes also include screening for congenital critical heart disease and for hearing disabilities; this study is limited to those conditions that are identified by screening using neonatal blood samples.

4) Table 2 column headings are entirely comprised of abbreviations which should probably be explained in a footnote. Failing that, a Table footnote should refer the reader to the list of abbreviations provided at the end of the text.

We agree with this reviewers opinion and added a note to the table 2 that refers to the list of abbreviations. 

5)  Figure 1 is difficult to read and lat/long scales should be added to the axes-the reviewer makes several suggestions to improve the figure.

We agree with this reviewer and provide a revised version of Figure 1. Instead of making the figure more complex by adding lat/long scales we coloured all countries not involved in this study in grey, which also highlight the shape of the Caspian and Black Sea.

6) Reviewer 3 requests a discussion of the question of the relative merits and/or a draft priority list for neonatal DBS screening for various conditions to assist policy makers faced by this wide variation in the current practice of neonatal DBS screening within Europe.

We feel that this point of reviewer 3 is already implicitly covered in par.3.4.4 of the manuscript. However, to stress this even more, we added two sentence in the concluding paragraph.

This paper should be regarded as …… among countries regarding the screening, diagnostic confirmation as well as fol-low-up should be clear.

Whether this will ultimately …., remains to be seen.

Round 2

Reviewer 3 Report

In this second submitted draft, the authors have taken steps to address some of the points in my review of the first submitted draft.

The editors of the IJNS will be very well placed to decide whether the title should reflect the fact that the paper is not about neonatal screening generally but is confined to the topic of screening on neonatal dried blood spots.

The presentation of Figure 1 remains, in my opinion, unclear, unnecessarily complex and confusing. It would help to state that it is a map. Ideally the type of map (e.g. Mercator projection) might be stated since this will materially affect the distortion of the size of the image relative to surface area of the mapped regions - very relevant to the high latitudes of Scandinavia and the Russian Federation. A simple alternative would be to add and label lines of Latitude and Longitude.

Deciphering these confusing images requires a significant investment of time. I doubt that many readers would be able correctly to identify which nations are which around the east/west divide between the Commonwealth of Independent States and points immediately west of that region or even which seas are shown where in the images. Two pairs of images are presented in two rows. the first row has a single (unexplained and in my view unnecessary) red horizontal line (of equal latitude) through the two images. The other pair - in the lower row - appears identical to the first pair - in the upper row - other than being about 1.3 times larger and lacking the horizontal line.

In each row, The left hand image is an enlargement of a portion of the the left hand half of the right image except that most of the enlarged area has been deleted from the right hand image.

More specifically, the right hand image in each pair is missing Finland, the Baltic States, Poland, Romania and all points West except Greece which is retained. The deletion of Western Europe makes it extremely difficult to understand how the left hand image maps on the right hand image particularly because the coast line around the Western half of the Black sea is missing from the right hand image and most of the Caspian sea is not visible in the left hand image. Consequently pattern recognition between the two images in each pair is very difficult.

The intention of the maps is simply to document the timeline which clearly sweeps from West to East. This could probably be easily presented with a single large image of WHO Europe (in landscape orientation) and colour coding much as shown, possibly with the some labelling of two or more of the Caspian, Black, Baltic and Mediterranean seas. Alternatively, it could be achieved with one pair of images (in landscape orientation), the left hand image being similar to the current lower left hand image and the right hand image similar to the current lower right hand image except that it would retain the entire coast line of Western Europe retained, with the addition of a box (or circle, like a magnifying glass)  around Western Europe. This would not have to coincide exactly to national boundaries as the appropriate colour coding could be shown on the right hand image for parts of the countries bordering but falling outside the box/circle  Within that box there would be no colour coding. An arrow would link that box to the left hand image which would show the colour coding for Western Europe. Just like a town plan with an enlargement of the city centre.

Neither the text nor the Figure Legend currently explain the current Figure 1 images but instead I strongly recommend simplifying Figure 1 as suggested and then adding a legend along the lines of : ' Figure 1. Map of Europe, colour coded by starting year of neonatal screening, with an enlargement of the Western European region.'

You could add to the legend or, better as part of the colour coding key, 'Arrow indicates Israel (yellow). Albania and Tajikistan (in blue-stippled white) have no neonatal screening programme.  Grey areas fall outside the geographical region under consideration. White areas are bodies of water.'

Author Response

We thank the referee for his comments.

We reconsidered changing the title as suggested by the referee. However, we feel that the concept of "neonatal screening" in general is considered to refer to metabolic screening in blood samples obtained from the heel or from the umbilical cord. This is supported by the results of scanning the literature. Hence, we decided not to extend the title. Of course, the IJNS editor may have a different view.

The main purpose of Figure 1 is only to give an impression of how NBS developed across the European region. It is not a geographically precise map. Readers with less knowledge concerning the individual countries can easily understand the intention. We maintain that putting longitudinal and latitudinal scales would unnecessarily complicate the figure.

However, to please the referee we have chosen for one figure that covers (almost) the whole geographical region discussed in this paper.  It is now clear what land masses and water areas are and we do hope that this is acceptable.
We also followed the referee's recommendation regarding the legend.